# Being Bayesian: Discussions from the Perspectives of Stakeholders and Hydrologists

**Ty P.A. Ferre**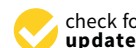

Department of Hydrology and Atmospheric Sciences, University of Arizona, Tucson, AZ 85721, USA;
tyferre@email.arizona.edu

**Abstract:** Bayes' Theorem is gaining acceptance in hydrology, but it is still far from standard practice to cast hydrologic analyses in a Bayesian context—especially in the realm of hydrologic practice. Three short discussions are presented to encourage more complete adoption of a Bayesian approach. The first, aimed at a stakeholder audience, seeks to explain that an informal Bayesian analysis is the default approach that we all take to any decision made under uncertainty. The second, aimed at a general hydrologist audience, seeks to establish multi-model approaches as the natural choice for Bayesian hydrologic analysis. The goal of this discussion is to provide a bridge from the stakeholder's natural approach to a more formal, quantitative Bayesian analysis. The third discussion is targeted to a more advanced hydrologist audience, suggesting that some elements of hydrologic practice do not yet reflect a Bayesian philosophy. In particular, an example is given that puts Bayes Theory to work to identify optimal observation sets before data are collected.

**Keywords:** hydrology; modeling; decision-support; uncertainty; measurement optimization

## 1. Introduction

A discussion of the application of Bayes' Theorem for decision support should probably start with a definition of Bayesian analysis. Simply stated, Bayes' Theorem is a guide to updating the relative probabilities of competing hypotheses as more information becomes available. The term 'updating' is key: probabilities are not simply calculated anew when new information is introduced; rather, the new information is interpreted in the context of the previous data and analyses. This allows for a smooth transition from the current belief toward a new interpretation. But an approach based on updating must start with set of hypothesis and initial estimates of their relative probabilities. Bayes allows these initial probabilities to be based on a combination of subjective opinion and objective data. This inclusion of subjectivity can be seen as a weakness of Bayesian analysis. Through the following three discussions, a case will be made that allowance for subjectivity is actually a strength of Bayesian analysis for decision support and that the subjectivity goes deeper than the choice of the initial hypotheses to be tested.

For those of us who spend time thinking about how hydrologic models can and, perhaps, should be used to support water resources decision making e.g., [1–19], the introduction of Bayes' Theorem into hydrology marked a major turning point. There is little argument that Bayesian analysis is on the rise in hydrologic circles [4,20–36]—with the number of publications returned from a Web of Science 'Bayes hydrology' search languishing at fewer than 10 until 2009, then increasing dramatically since (Figure 1). Of course, the number of hydrology papers has increased in this same time, as well. So, despite the rapid recent increase in Bayes-related citations, far fewer than 1% of hydrology papers include a reference to Bayes' Theorem. The rise in hydrologic reference to Bayes' Theorem and the still rare inclusion of Bayes in hydrologic papers overall, suggests that it may be useful to provide a simple primer on what Bayes means for practicing hydrologists and the decision makers that they inform.

What follows is not meant to be a comprehensive, mathematical treatise on Bayes. Rather, it is meant to provide a first introduction to those who do not know anything about Bayes' Theorem or, at least, who hope never to be asked publicly if they are a Bayesian. (Short answer yes, you are.) This discussion is also meant to challenge practicing hydrologists to put their Bayesianism to work more effectively. Specifically, applied hydrology should focus on the effective use of hydrologic models for decision support, which relies on the thoughtful integration of measurements and models: measurements must constrain model-based interpretations and models should guide the design of measurement campaigns.

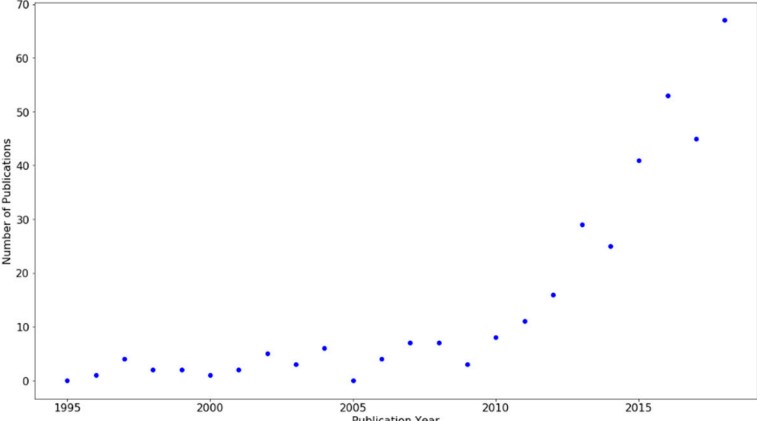

**Figure 1.** Number of publications returned for the search of 'Bayes hydrology' in Web of Science as of December 2019.

This discussion is divided into three parts. First, a Bayesian analysis of a water resources problem is presented for non-experts. Then, a discussion is provided to link the qualitative Bayesianism that underlies everyday decisions to a more quantitative application of Bayes' Theorem by to water-resources decisions. Finally, the concept of Bayesian analysis is extended into hydrologic measurement design as part of a challenge to hydrologists to embrace Bayesian analysis more completely.

## 2. Bayes from a Stakeholder Perspective

Whether you know it or not, you are a probably a Bayesian. Consider a simple example of everyday decision making. Getting ready to leave the house, you look out the window and see clouds gathering. Should you wear a rain jacket? If you are wearing a nice outfit and you do not find a rain jacket too cumbersome, then there is little risk in wearing one—you spend very little time and effort making the decision because an in-depth analysis is unlikely to change your mind. Similarly, if you are a young student, for whom the perceived social cost of wearing a rain jacket is so high that you would not do so even if there was an epic rainstorm occurring at that moment, the decision to go without a jacket is simple and immediate. People in these extreme conditions do not need to be Bayesian because they do not have to make a considered decision. But what if the trade-off between cost and benefit is not immediately clear? How do you decide if the probability of getting caught in the rain without a jacket is high enough to outweigh the inconvenience of bringing one? The first sign that you are a Bayesian is that you have cast the problem as a probability. The second sign, even though it may not be obvious, is that you have already made an estimate of the probability. It is almost certainly a poorly informed and not particularly accurate estimate: for example, it has been well documented and well explained that this probability estimate is subjective and highly influenced by personal biases [37]. But regardless of its accuracy, you have at least decided that the perceived probability is high enough in the context of the risk and cost that it is worth seeking more information. The path from pure reliance on experience or rules of thumb to a more thoughtful analysis is through the most available data, perhaps using a weather app. If the app makes what appears to be a very certain forecast (0% or 100% chance of rain), you may be willing to accept it in place of your initial

assessment. If you were to reason it through, it would be hard to imagine how the forecast could be 0% (100%) and yet you would (not) be rained on. But if the forecast is only 20%, then you would be able to craft many possible stories that would end with you being rained on or not. If your biased initial estimate led you to doubt that it would rain, you might see this as confirmation and accept it. If your biases led you to think that there would be rain, you could easily discount this forecast. That is, you view the information in the context of your previous conclusions informed by the data that you had available before opening the app. If you want even more certainty, you would seek information that would test those competing storylines. Perhaps you have noticed that you get rain in your part of the city more often when the storm approaches from the west. Scrolling through the app, you find the weather radar, which does show the storm approaching from the west. You do not discard your initial assessment, or the assessment finding that there was a 20% chance of rain. Rather, you update it, nudging it toward a threshold probability that would lead you to bring a jacket. This is the second sign that you are a Bayesian: you continually update your probability estimate by interpreting new data in the context of the question being asked and the conclusions that you had drawn before collecting the additional information. In fact, this common place approach is not just Bayesian, it is an advanced form of Bayesianism in which you intentionally sought information that could specifically challenge the probabilities of the storylines that were driving your decision. You looked for additional data that was most relevant in the context of your threshold for action (bringing a jacket) and the information and storylines that you had in mind.

When dealing with non-scientists, it is useful to remember that hydrologists are people, too. We make thousands of decisions every day and many, if not most, of them that require any thought will follow a similar Bayesian path. Based on our common humanity, it should be relatively simple to describe a hydrologic analysis for decision support as Bayesian. It may even be possible for hydrologists to improve their professional analyses by referring to their everyday decision-making skills. Let's try for a simple water resources problem.

Consider a person who lives in a relatively undeveloped watershed. A new business has applied for permission to pump ground water. The local resident's primary concern is that the pumping will impact the flow in a local stream, with associated negative impacts on the environment and recreational use of the water body. The resident is not entirely opposed to the new development; they want to better understand the possible impacts on the local riparian system before deciding whether to support or oppose it.

Upon hearing about the proposed pumping, the local citizen has a bad feeling: if they had to put a number on it, they would say that they were 75% sure that the pumping would damage the stream. This probability is, essentially, a fiction. It represents the stakeholder's concern, amplified by risk aversion, much more than an assessment of the probability of a negative outcome. Mirroring the jacket decision, the first step forward is to determine if the stakeholder can be influenced by new information. If, like a young person who adamantly refuses to wear a jacket in a rainstorm, the stakeholder is completely resistant to changing their mind about supporting the new water use on the basis of a hydrologic analysis, then the hydrologist should carefully consider whether the effort of such a study is worthwhile (beyond the fee that they will charge to complete it). If the stakeholder is open to scientific guidance, then they have to describe the tradeoff decision as they see it. For example, they may set a threshold minimum stream flow or maximum fraction of loss of stream flow that is acceptable. Or, in some cases, they may be able to describe a continuous relation between loss of stream flow and their level of acceptance of the outcome. Note that, in this case the critical hydrologic outcome is loss of stream flow, but any impact could be considered: reduction in baseflow, number of days of flow, average flow, even water temperature during a spawning season. Regardless, it is critical to recognize that the stakeholder must relate the hydrologic forecast(s) that hydrologists can provide to their satisfaction, which only they can define. Researchers from many fields have contributed to the idea of valuing ecosystems, including hydrologic impacts on ecosystem worth [38–42]. But at some level this valuation is, fundamentally, subjective.

At this point, the stage is set for a Bayesian hydrologic analysis. The stakeholder would like to find a ready source of information to improve their estimate of the probability of damage due to the proposed pumping. Without resorting to hiring a hydrologist, the stakeholder could simply search for other watersheds that have faced similar added pumping to see how they fared. Assume that they managed to find 20 other examples. Of these, four had what they would deem unacceptable impacts on the riparian system and 16 did not. As for the jacket decision, if you initially supported development, then this 20% may be a clear confirmation of your decision. But a stakeholder who initially thought that the chance of damage was very high is less likely to discard their opinion on the basis of this new data. They may argue (correctly) that the percent of other basins that saw damage has little or nothing to do with the probability that their basin will experience negative impacts. In fact, it is a leap of faith to assume that the collection of other basins can be used to forecast future conditions in the stakeholder's basin. This belief requires that at least one of the basins that we found is similar enough to our basin to act as a reliable surrogate; more fundamentally, it requires that we have some way to assess basin similarity that can be measured now, before pumping begins and (potentially) impacts the stream. This is different than the jacket problem, for which we could turn to an app that was specifically designed to make the prediction of interest and has been trained using a lot of historical weather data. The analog basins are less reliable than a weather app because they are relatively rare, and it is difficult to determine how similar they are to the stakeholder's basin. As a result, so they require more consideration to assess their information in the context of the question being addressed and the analyses preceding data collection.

What is the hydrologic equivalent of identifying the storm direction for the jacket problem? In other words, what can we measure now that categorizes the analog basins as more- or less-similar to the stakeholder basin in the context of pumping impacts? That will be addressed in the next discussion; for now, we can only ask the stakeholder to trust that we will provide an explanation. Then we can ask how we would use a measure of similarity. For simplicity, imagine that we can show that two of the surrogate basins are very similar to the stakeholder's basin and the other 18 are less similar. Of the two more-similar basins, one saw unacceptable damage. Which probability should the stakeholder accept? The initial 75% probability? Or, the 20% assessment based on all analog basins? Or, the 50% likelihood of damage based only on the more-similar basins? There is no right answer, but there is a best practice. That is, you can adopt the 75% probability if you can defend the argument that the analog basins provide no relevant information. You can believe the 20% probability if you can defend the proposition that the 20 basins are collectively representative of the stakeholder's basin and that they are all equally informative regarding the fate of the stakeholder's basin. Or, you can use the 50% probability if you can show that the more-similar basins represent the stakeholder's basin so much better, that the less-similar basins can be ignored. The best practice is to state the underlying assumptions and then calculate a probability of damage that is consistent with those assumptions. Different stakeholder groups can then debate these probabilities and look for further information to refine them if such refinement could change their decisions.

Bayes' Theorem is a guide to incorporating new data to update an estimate of the probabilities of competing hypotheses. This can be shown graphically for the stakeholder's water resources problem. Assume that there is something that can be measured in a basin that can define how similar that basin is to the stakeholder's basin. To begin, imagine that the 20 surrogate basins are represented as points that fill the white square in Figure 2. Of these 20 surrogate basins, four saw damage. Those basins are enclosed in the blue ellipse, which represents 20% of the area of the white square. There were two basins that were found to be most-similar to the stakeholder's basin. These more-similar basins are gathered in the red ellipse which fills 10% of the white square. Finally, one of the more-similar basins saw damage, so half of the red ellipse overlaps with the blue ellipse.

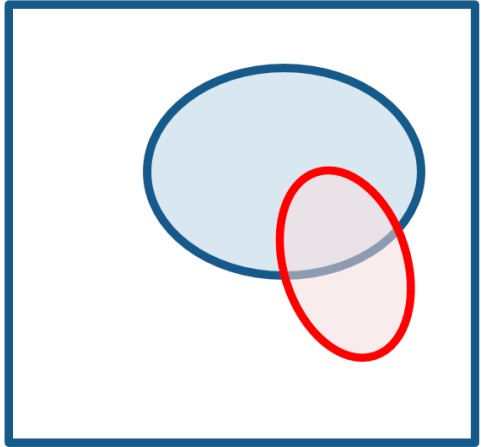

**Figure 2.** Illustration of joint probabilities as the basis of Bayes' Theorem. The white box contains all models in the ensemble. Models within the blue ellipse forecast unacceptable damage due to pumping. Models in the red ellipse are deemed more-similar to the stakeholder's basin.

The next step requires a small amount of simple math. First, we give variable names to the probabilities that have already been illustrated. The probability, $Pr$, that any basin saw damage, $D$, is: $Pr(D) = 0.10$. The probability that any basin was similar to the real basin was: $Pr(S) = 0.20$. We can also describe the probability of one thing happening if another is already known to be true. For instance, given that a surrogate basin lies within the red ellipse, what are the chances that it also lies in the blue ellipse? This is written, $Pr(D|S)$. Half of the more-similar basins also had damage. Or, $0.5\ Pr(S) = Pr(D|S)$. Finally, of the four basins that saw damage, only one was a more similar basin. That is, $0.25\ Pr(D) = Pr(S|D)$. Both $Pr(D|S)$ and $Pr(S|D)$ are the area of the overlapping region of the two ellipses divided by the area of the white box. Because they are identical, we can state:

$$Pr(D|S)Pr(S) = Pr(S|D)Pr(D) = 0.05 \tag{1}$$

Rearranging gives Bayes' Theorem:

$$Pr(D|S) = Pr(D)\frac{Pr(S|D)}{Pr(S)} \tag{2}$$

The fraction, $Pr(S|D)/Pr(S)$ is referred to as the evidence, E. In this case, it is a measure of how much more we should rely on a more-similar basin than a less-similar basin in the determination of the likelihood of damage due to pumping. At one extreme, we may believe that the similarity measure has no value, or $Pr(D) = Pr(D|S)$. Thus, E = 1. At the other extreme, we may believe that we should only consider the more-similar basins, $Pr(D|S) = 0.5$. This gives E = 2.5. If we can defend some level of imperfect information in the similarity measure, then E will be value between 1 and 2.5. If the stakeholder is not a hydrologist, they are unlikely to know what measures are available to assess basin similarity or how to define the evidence provided by these measures. But this explanation may give them insight into what a hydrologist is doing on their behalf. What is more difficult to see from this representation of Bayes' Theorem is that the key steps to the analysis are: which basins have been included in the set of surrogates, and which information has been considered to assess similarity as it applies to the decision. Once these decisions are made, then it is relatively straightforward to apply Bayes' Theorem. Unfortunately, many advanced investigations that apply Bayes' Theorem give little or no guidance or justification for these two critical choices. The next two discussions aim to address each of these topics in turn.

To summarize the stakeholder discussion, a case can be made that all of us follow a common path to many of our daily decisions. The first step is to imagine the range of possible outcomes stemming

from the decision. Then, we determine the costs of the possible outcomes. If the costs are uniformly high or low, then the decision may be obvious with little or no further consideration. If the cost versus benefit does not lead to an immediate decision, then we cast the problem as a probability. The initial probability estimations are likely inaccurate and biased. If we have enough faith in them to make a decision, then we move on. But if we have a nagging uncertainty, then we seek additional relevant information. Once we collect new data, we revise our probabilities. But it is rare that we have absolute confidence in the implications of the data for the decision. We need some context to decide if the data add further support to, or detract from, one or more possible courses of action. We continue this process until we are confident in the probabilities driving the decision, or until we run out of time or money. Ideally, we make this process as efficient as possible by seeking the information that is most able to influence our decision.

## 3. Bayes from a Hydrologist's Perspective—The Basic Story

There has been considerable interest in the process of decision making and its interaction with scientific information, also including hydrologic applications [4,37,43–45]. This discussion focuses directly on bridging from the general description of a Bayesian decision approach, as presented for stakeholders, to a more quantitative description of a hydrologic investigation. Emphasis is placed on the question of how to formulate the equivalent to a collection of surrogate basins, and how to identify observations that are most likely to change our assessment of the relatively similarity of these basins to the stakeholder's basin.

A Bayesian decision approach is a restatement of the scientific method: produce competing hypotheses; identify the data that are most likely to test the hypotheses against one another; collect the data and use it to provide evidence in support of some hypotheses over others. Beyond formulating the competing hypotheses, the most difficult elements of this process are: assessing the relative reliability of different sources of information and identifying additional sources of information that are most likely to impact decision-relevant forecasts. This discussion is aimed at putting these two challenges in a hydrologic context.

The first difference between the stakeholder example and a typical hydrologic analysis is that, as hydrologists, we rarely have enough case studies to rely on surrogate basins: case studies are expensive and ours is not a wealthy science. Rather, we rely on mathematical models to act as virtual case studies. It is worth accepting and admitting that models are not reality, even if we have worked very hard to build them and have used all of our considerable skill to achieve a very low mismatch with the available data. Invariably, models leave out processes and/or structural details that could have bearing on forecasts, thereby affecting their value for decision support. As a result, model building requires care and competence both to get the models as right as possible and, as important, to explore those elements of the model that we know to be uncertain.

Model building should not be seen as an effort to build the best model of a site; rather, it should be viewed as an effort to build a collection of models that represent competing hypotheses. To the degree possible, a single model should not be used for decision making; decisions should be informed by a model ensemble. Intuitively, a stakeholder would understand the value of considering multiple analogs rather than relying on one surrogate basin, even if they could be assured that it is more similar to their basin than the other surrogates that they happened to find. Perhaps this is why there is resistance to believing a single mathematical model, regardless of how carefully it has been crafted. As an aside, there can be a tendency to get mired in the definition of *model* in such a statement. For this conversation, the widest possible definition is adopted: any changes to parameter values, model structure, boundary conditions, or forcings constitute a new model. In the context of the analog basins, if a stakeholder found two different basins that only differed in climate, or soil type, or applied hydrologic stress, they would be considered to be two different basins for the purpose of decision support. The relatively low cost of models (compared to case studies) is a key strength of model-based analyses. We can afford to construct many models of a system and we can intentionally explore conditions that may or may

not be found in accessible, existing analog basins. This is a subtle but important distinction. When seeking surrogate basins, we are limited by what exists with sufficient supporting information. When building models, we have the freedom to construct many models that could, plausibly represent the conditions in the stakeholder's basin. With this freedom comes the responsibility to seek out models that matter. That is, the ensemble should include as many plausible models that could profoundly affect the stakeholder's decision as possible. Given that many stakeholders will display risk aversion, this equates to a requirement that a hydrologist should include as many models that forecast a low utility outcome (based on the stakeholder's valuation) as possible. In other words, if a stakeholder is biased toward rejecting a proposal on the basis of future damage to the environment, then the most valuable service that a hydrologist can provide is to examine and, potentially, discount as many pathways to this damage as possible. This will almost certainly be more valuable than providing a single model, with all of its inherent assumptions, that suggests that there will be no negative impacts. In this context, it should be clear that an effective model ensemble is not simply a large number of models. Creating an ensemble of models to intentionally examine uncertain and decision-critical questions is a unique modeling skill that is not an automatic outgrowth of the skills needed to build a single, well-calibrated model. The remainder of this article presents some key insights necessary to conduct a multi-model hydrologic analysis in a Bayesian context.

For the stakeholder example, the foundation of a successful analysis is the collection of analog case studies; the rest of the Bayesian analysis focused on revising the relative reliance on these sources of information. For a Bayesian hydrologic modeler, the foundational step is to form a useful model ensemble. The simplest view of a model ensemble is that each model represents a competing hypothesis about the hydrologic functioning of the system. In addition, taken as a whole, the model ensemble should represent what we know (e.g., physical laws), what we think we know (our prevailing assumptions regarding structures and relevant processes), and what we know that we do not know (less certain assumptions regarding structure, processes, and parameter values). These prescriptions for forming useful models and model ensembles have been discussed previously [2,4,46–52]. Similarly, the element of surprise [53,54], which describes the discovery of unknown unknowns, represents the collection of elements that were not included in any model in the ensemble. The art of modeling relies on the ability to identify a model that best matches all available data while obeying known physical laws and, often, while requiring the least complexity. The art of ensemble modeling includes these skills, but it also requires that multiple different plausible, behavioral models be constructed and that these models cover all decision-relevant unknowns.

Thoughtful formation of an ensemble of models allows us to explore the system while minimizing the impact of specific choices embedded in any one model. Ensemble modeling also allows us to define and describe many possible sources of uncertainty, rather than focusing on uncertainty related to small variations of a single best-fit model. But this comes at a cost. Namely, if the same computational effort is expended on a running a model ensemble as is spent on calibrating a single model, then the best-fit model in the ensemble is unlikely to be as well tuned as the single best-fit model. Two comments are warranted on this point. First, it is worth deciding if a marginally better fit to limited data truly constitutes a better model for decision support. Second, the computational demands of ensemble modeling may provide further support for preferring less complex models, as has been discussed elsewhere [10,14,29,46,48,52,54–60].

The stakeholder discussion concluded with the idea that not all models are equally important for decision support. Rather, it is critical that the model ensemble includes, and perhaps even focuses on, models that are most consequential. The consequence of a hydrologic forecast is defined based on stakeholders' subjective utility functions. This level of subjectivity may make some modelers uncomfortable [18,61]. But modeling represents a compromise; no model is optimal for all applications. Therefore, it seems sensible to focus model-building efforts on improving performance related to outcomes that matter for decision support. Decisions often hang on assessing the likelihood of low

probability, high impact models. This can only be done effectively by comparing these models to other models as part of an ensemble.

As presented to the stakeholders, and at the risk of stating the obvious, hydrologic models can only make hydrology-related forecasts. As a result, the first step in modeling for decision support must be the definition of the problem(s) and the second must be translation of these problems to decision-relevant hydrologic forecasts. Furthermore, these two steps should be made by each stakeholder (group). For example, the stakeholder described in the previous section would define the problem as decreased recreational benefit of the stream. This is related to decreased streamflow due to the proposed pumping. Then, the level of decreased flow is related to their relative satisfaction, possibly adding a threshold of acceptability if available. A different stakeholder, with different primary concerns or interests, would formulate a different problem that may require different hydrologic forecasts. A model ensemble should attempt to accommodate all of the forecasts that drive all of the stakeholders' decisions, with a focus on proposing plausible models that could change each stakeholder's mind. A continuous dependence of a stakeholder's satisfaction on a hydrologic forecast is referred to as a utility function; an example is shown in Figure 3. The process of constructing a quantitative relationship between utility (relative preference) and hydrologic outcome is contentious and involved. For this discussion assume that some form of a utility function can be constructed to describe a stakeholder's preference for different outcomes. After all, if such a relationship cannot be described, then it is unlikely that the results of a hydrologic analysis will be particularly useful for decision support.

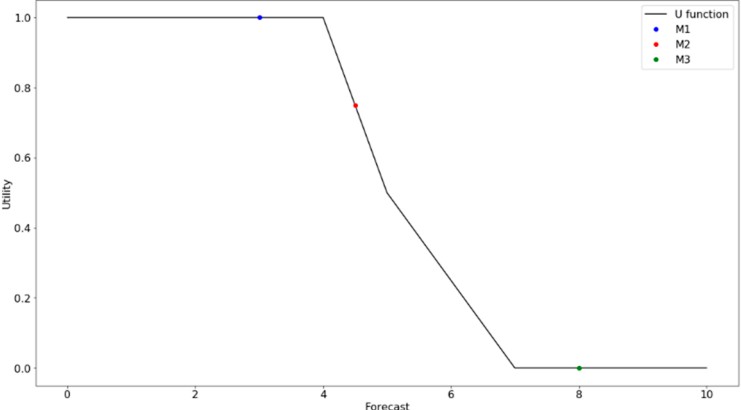

**Figure 3.** Stakeholder-defined utility as a function of a key hydrologic forecast. The forecast utility is shown as points for three models: *M1*; *M2*; and *M3*.

For the remainder of this discussion, we will consider a very small ensemble comprised of three models: *M1*, *M2*, and *M3*. Three points are shown on the utility function (Figure 2), one representing the stakeholder-defined utility associated with each models' forecast streamflow reduction due to pumping. Here, the streamflow reduction forecasts made by models *M1*, *M2*, and *M3* have associated utilities of 1.0, 0.75, and 0, respectively. This could be reduced further to acceptable or not acceptable by assigning a threshold of acceptability. For example, for the utility function shown, if the required utility threshold for a stakeholder to support the proposed pumping was 0.8, then 1/3 of the models would forecast unacceptable impacts of pumping.

The next step in the stakeholder's Bayesian analysis was to classify analog basins as more- or less-similar to the real basin. The binary 'acceptable' or 'unacceptable' classification is equivalent to stating that a model makes a prediction that leads to a utility that is above or below the stakeholder-defined threshold. The utility function replaces this binary valuation with a semi-quantitative, continuous measure of utility. If we can also replace the binary more- and less-similar

classification with a continuous probability, then we can calculate a probability-weighted utility, $\overline{U}$, from the probability, *Pr*, of each model and the utility, *U*, determined from the forecast(s) of each model:

$$\overline{U} = Pr(M1)U(M1) + Pr(M2)U(M2) + Pr(M3)U(M3) \tag{3}$$

This is slightly different than the development of Bayes' Theorem following Figure 2. There, the probability was summed over models based on their sharing a class of forecast (acceptable or not acceptable). The approach described here is referred to as Bayesian model averaging—it still relies on the association of model likelihood based on one set of measures to quantify the reliability of a model for different forecasts. But it has the clear advantage that a model that is close to the threshold of unacceptable or similar is not lumped with clearly unacceptable or clearly similar models. More generally, this continuous approach allows us to fall on the spectrum between ignoring model similarity and relying exclusively on most-similar models. But the same caveats apply to a continuous probability as to classification of more/less similar: the choice of measure of model probability should consider all decision-relevant characteristics, it should be as closely related to the basis of the decision as possible, and it will ultimately be subjective. The default, if there is no basis upon which to assess the model probabilities, is to apply a 'uniform' weight to all of the model forecasts. This is equivalent to the extreme case of considering all analog basins to be equally informative, regardless of their similarity to the stakeholder's basin. This is a common starting point of Bayesian analysis which is referred to as assuming an uninformed (or uniform) prior probability distribution.

Having made the case that model likelihoods (probabilities normalized to sum to one over all models) can be continuous, it is reasonable to ask how model likelihood is calculated. Hydrogeologists assess model likelihood in two ways. If a model is not consistent with accepted physical principles, then it is deemed implausible and excluded from consideration. Plausible models are deemed more or less likely (to be true or reliable or representative ... it is rarely stated) based on their goodness of fit to all existing data. In some cases, models are further penalized for added complexity [62–66]. There are many possible measures of likelihood, each preferred for different applications, each with its own vocal supporters. They can all be understood in the context of Bayes' Theorem through a simple example. Consider the three models (*M1*, *M2*, and *M3*) and three potential observations (O1, O2, and O3). We need a continuous measure of the probability that a model forecast of a measurement agrees with the observed value. Conceptually, this continuum definition balances two insights. First, given that all models are simplifications of reality and that measurements are imperfect, it is unreasonable to expect that a model forecast will match an observation perfectly. In fact, a perfect match should be discounted somewhat because it is, in part, a fluke of cancelling errors. At the same time, a closer match between a forecast and a corresponding observation should provide evidence in support of the model making the forecast. (If not, then what is the value of collecting data for assessing models?) One way to satisfy both of these requirements is to describe the probability that any model's forecast agrees with the observed value using, for example, a Gaussian (normal) error with zero mean about the observed value with a variance that represents a combination of the true measurement error and limitations due to model simplification. Of course, a different error model could be used if it can be justified – the choice is, to some degree, subjective. To illustrate a Gaussian error analysis, consider the three observations (O1, O2, and O3) with measured values of 6.15, 4.00, and 4.75 variances of 1.0, 2.0, and 0.75 (Figure 4). Probability distributions around the three candidate observations are compared to the corresponding forecast of that observation made by each model (without error). In this case, the measured value of O1 would provide evidence for *M1* at the expense of *M2* and *M3*. O3 would provide evidence for *M2*. But O2 would not discriminate strongly among the three models. All measures of model/data mismatch follow a similar approach; higher mismatch is penalized and the rate of change of the penalty decreases the greater the mismatch. The approach illustrated here is the closest expression of the probabilistic nature of Bayes' Theorem; but it is often replaced with less computationally expensive measures such as the root mean squared error.

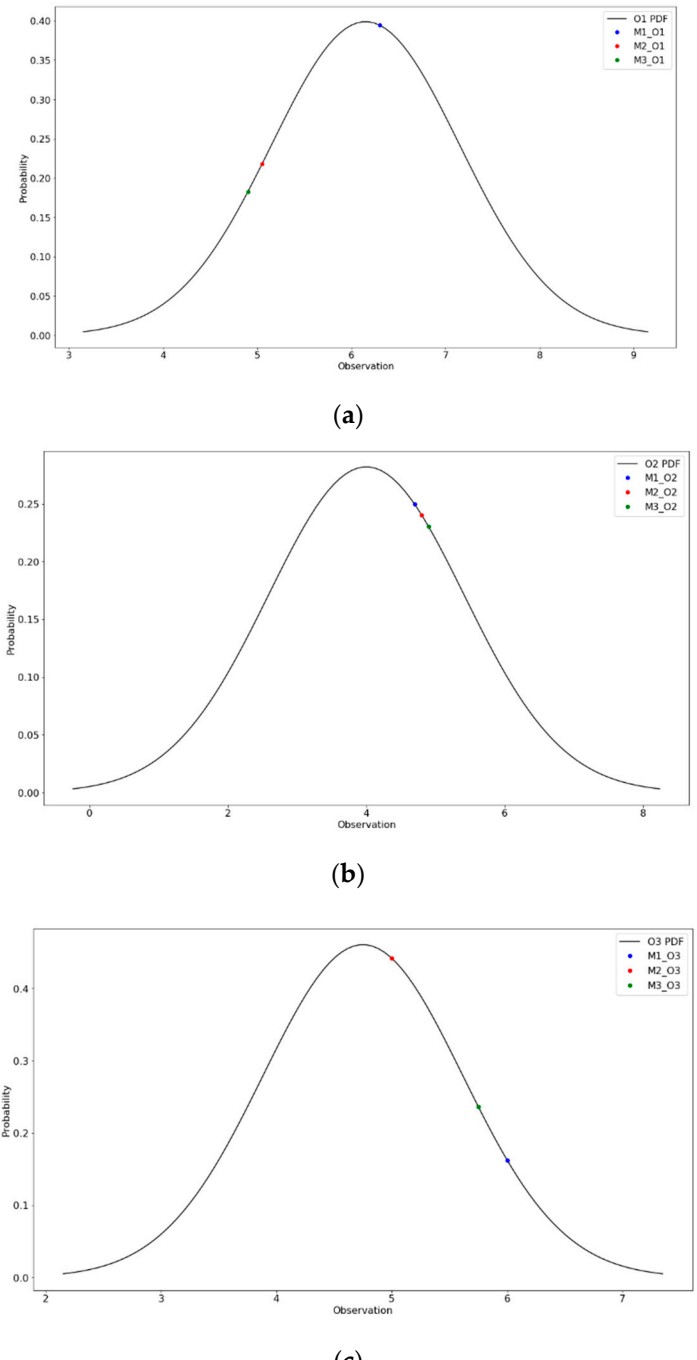

**Figure 4.** Probability distribution based on observed value and assumed variance with corresponding model-forecast observations for three proposed observations: (**a**) O1; (**b**) O2; and (**c**) O3.

This discussion of model/measurement mismatch as the basis for assessing model likelihood leads naturally to consideration of the optimal choice of observations for a hydrologic investigation. Because data collection often represents a major cost in hydrologic investigations, this topic has received considerable attention in the literature [4,67–86]. Arguably, measurement optimization is still used sparingly in practice.

Even before the observations shown in Figure 4 have been collected, one conclusion may be immediately apparent: it is unlikely that O2 is worth collecting because all three models predict very similar values compared to the measurement uncertainty (variance). That is, try to ignore where each point lies with respect to the peak of each black probability curve on Figure 4. O2 and only consider

how different the forecasted observations are compared to the variance applied to the observation. Observation O2 has very little ability to discriminate among the models—or, collecting O2 is unlikely to change the relative probabilities of the three models. It is also evident, before collecting the data, that O1 is likely to discriminate model *M1* from *M2* and *M3* whereas O3 is likely to discriminate *M2* from *M1* and *M3*. But it is difficult to see the potential value of these distinctions for decision support. Once the observations have been collected, a post-audit can be performed to calculate which observation had the greatest influence on the probability-weighted forecast and associated utility. Such a post-audit can be extremely valuable for honing intuition regarding the value of observations. In fact, it can be argued that the almost complete lack of post audit analysis in hydrologic science and practice is a tragic missed opportunity; but it is one that is unlikely to be addressed given the current funding models of academic research and consulting practice, which commit very limited funds to reanalysis of existing data. Regardless, for a study in progress, there is little to be gained from a post-audit analysis because the time and effort have already been expended to collect the data.

One approach to analyzing the potential value of data before collecting it is to examine the value of an observation over a range of plausible observed values. In other words, we may not know what we will measure for O1, but we can assess how valuable O1 would be in the context of the Bayesian model average forecast (and associated utility) for a range of possible measured values. Specifically, we consider the evidence, E, for each model calculated for each observation over a range of possible measured values (Figure 5a–c). These results, determined before data are collected, show that a high observed value of O1 would support *M1* and decrease the likelihood of *M2* and *M3* (Figure 5a). But a low measured value of O1 would have more ambiguous results. Measurement O2 has little chance of providing strong evidence for any model (Figure 5b). Finally, O3 is similar to O1, but a low measured value discriminates *M2* from *M1* and *M3* while a high measured value is ambiguous. Finally, for each observation, there is a range of observed values that would result in very little evidence for any single model. The posterior probability-weighted likelihood of stream flow depletion due to pumping can be calculated based the E values at each assumed observed value. These forecasts can be translated to utilities through the function shown on Figure 3. Before the data are collected, the Bayesian model average utility is below the threshold of 0.8 (dotted black line on Figure 5d). This analysis shows that, regardless of the observed value of O2 or O3, the Bayesian model average utility will not exceed the threshold for the stakeholder to support the new development (dashed green line). From the point of view of the stakeholder, there is no value to collecting O2 or O3. In contrast, O1 has the possibility of changing the stakeholder's decision. Therefore, it is more meaningful for decision support, regardless of what is measured. One way to look at this is that choosing O2 or O3 imparts a bias in the decision process because neither can actually test the outcome in a way that is meaningful for decision support. This is an important point to understand and to communicate to stakeholders. There is no guarantee that any proposed observation will help to clarify their decisions; but it is possible to avoid choices of models to include in the ensemble and of data to collect that can lead to misleading analyses and poorly informed decisions.

Before examining more advanced concepts related to Bayesian hydrologic analysis, it may be helpful to summarize this discussion and relate it to the stakeholder discussion in plain language. The first step for both hydrologists and stakeholders is to define the problem and to relate it to hydrologic forecasts that will influence the decision. Then, the stakeholders must relate the forecasts to their preference in the form of a utility function. The details of this function are likely less important than the discussion that it will promote between the stakeholder and the modeler, with the modeler describing potential hydrologic outcomes that can be forecast and the stakeholder deciding which of these outcomes is more/less consequential to them. It is quite likely that each stakeholder (group) will have a different utility function that may consider different hydrologic forecasts. In a multi-stakeholder context, the construction of the model ensemble and the choice of optimal data to collect should be guided by identifying models that make impactful forecasts and observations that are potentially useful for testing those more consequential models. Once the stakeholders have defined the problem

in hydrologic terms, the hydrologist builds a collection of models that conform to physical principles, honor what is known about the system, and capture the range of what is (known to be) unknown about the system. In particular, the ensemble should include plausible models of the system that lead to different values of the decision-relevant forecasts. Data serve to define the similarity of each model to the stakeholder's watershed; these similarity measures are used to weight model predictions, potentially leading stakeholders to draw different conclusions based on the same data set. That is, it is a fallacy to believe that the data speak for themselves; Bayesian analysis is based on the idea that the context is all-important for determining how to view the data. With few exceptions, model similarity is assessed based on comparing data collected within the basin to the predicted values of those observations made by each model in the ensemble. The choice of the relationship used to calculate this goodness of fit is largely subjective; that ambiguity is consistent with the nature of a Bayesian analysis. Finally, additional data should be selected based on the possibility that it could change the stakeholder's decision. That is, the data must have the ability to change the weights on models in the ensemble such that the probability-weighted, Bayesian model average forecast utility is changed. There is little value for decision support in collecting data that have no possibility to inform (that is, change) the decision.

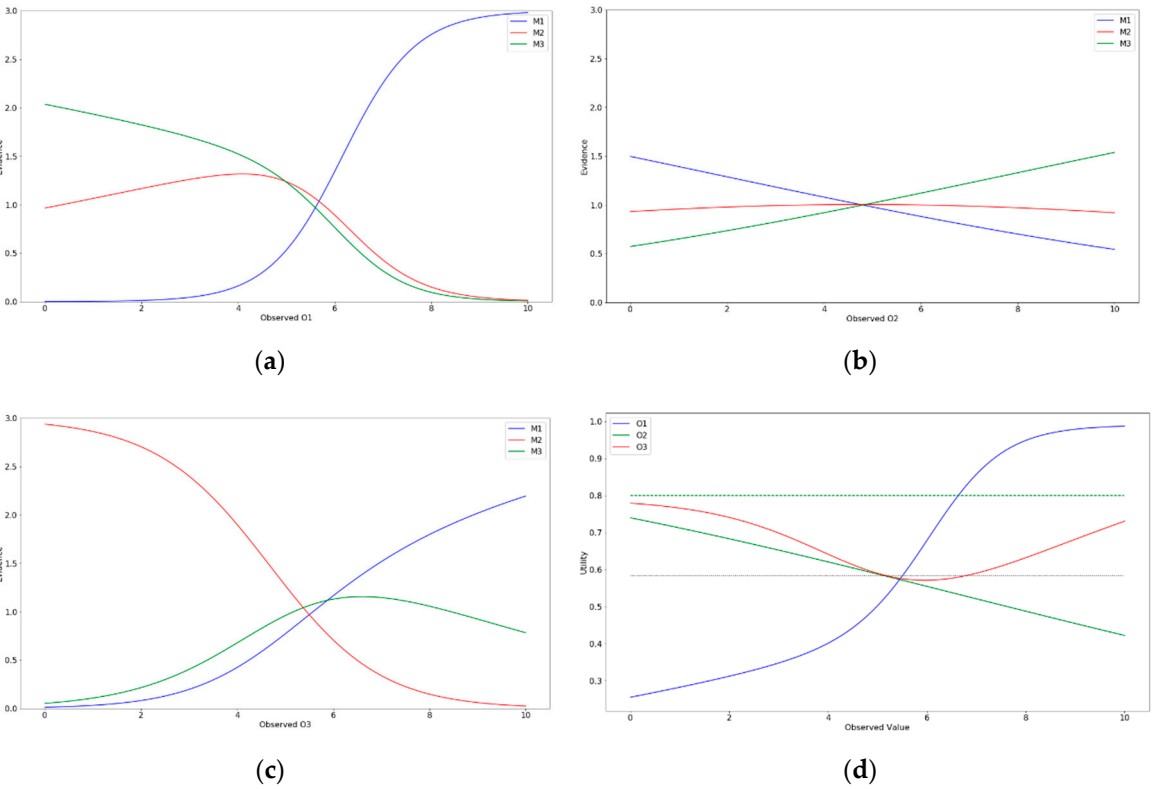

**Figure 5.** Evidence for each model (*M1*, *M2*, and *M3*) based on observed value of: (**a**) O1; (**b**) O2; (**c**) O3; (**d**) probability-weighted forecast utility as a function of observed value for each individual observation. Dashed lines: average utility before data collection (black) and minimum acceptable utility (green).

## 4. Bayes from a Hydrologist's Perspective—The Next Level

Beyond the summary provided at the end of the previous discussion, there are some fine points that may help a beginning practicing hydrologist to understand and adopt Bayesian analysis. These are presented as an introduction to the final discussion, aimed at hydrologists with more experience in Bayesian analysis.

The stakeholder discussion made that case that we all make everyday decisions that are essentially Bayesian. For a hydrologist, it is important to be more specific in defining what constitutes a Bayesian

analysis. Bayesian analysis is commonly contrasted with a frequentist approach. The latter is a relatively strict interpretation of probabilities as the long-term statistics of repeatable events. A Bayesian approach allows for probabilities to be assigned more broadly, including to parameters or hypotheses. Subsurface hydrologic studies rarely have enough data to construct probabilities and often target outcomes that have not yet occurred. So, if we want to supply decision makers with insights to inform probabilistic decisions, a Bayesian path seems well advised if not inevitable. If we accept that we want to perform a Bayesian analysis, the generic description given for stakeholders suffices with a few additions.

We cannot assess the probability that any model is true. We can fall back on the truism that all models are wrong, but that is not particularly helpful for decision support. Rather, we are testing models relative to other models. Then, we convert these probabilities to likelihoods by assuming that the probabilities (must) sum to one. This allows us to use an ensemble of models to make likelihood-weighted decisions. But it explicitly ignores the fact that we know that our model ensemble is incomplete; in fact, we have no guarantee that our ensemble includes any acceptably good models for forecasting future outcomes. That is, we have no guarantee that our ensemble, as a collective whole, considers all relevant uncertainties about the system. A similar limitation to an ensemble-based analysis is a lack of exclusiveness of the models. Include many models that are very similar to one another is equivalent to over-weighting one such model. This is a particularly difficult problem to address and it has been discussed in some detail by previous authors [87,88]. The best that we can do is to try to construct robust ensembles of models and recognize that our probabilities are, themselves, approximate and potentially subjective.

We ignore biases in our objective measures. Another requirement of a Bayesian analysis is that we apply objective measures of evidence when updating from the prior probabilities to the posterior. However, there is no single measure of goodness of fit that is universally preferred. Even if there is a preferred method for a specific application, there is no single approach to weighting measurements of different type or quality or measurements made at different locations or times to ensure that the model similarity gives a measure of the reliability of forecasts of interest. Rather, we rely on unstated or poorly defended preferences for similarity measures and objective function weighting schemes that almost certainly carry bias. We generally assume that a model that is, by some selected measure, better at matching existing data will be better at forecasting the future; but we make this claim without proof. Again, the best that we can do is to state our similarity measures clearly, present and defend our choices for weights in the objective function, and recognize that our probabilities are, themselves, conditional on many decisions made during the analysis.

Not all information is equally informative. Standard hydrologic practice begins with data collection and ends with model calibration. It is rare, sometimes even seen as unethical, to consider a stakeholder's utility function when constructing a model ensemble. But without this consideration, we are not building models that are most reliable in the range of forecasts that are most important for decision support. This is another expression of the idea that a model that is good for one thing (e.g., calibration) is good for everything (e.g., forecasting). In fact, it is probably more accurate to assume that the only way to make a general model is to make a generally poor model. One way to stay true to Bayesian concepts while providing better decision support is to explicitly include models that make forecasts that are most important consequences to stakeholders as part of a large model ensemble. These can be referred to as advocacy models because they advocate for the stakeholder's concerns. Given that the probabilities produced from a Bayesian analysis are approximate and possibly subjective, and that models that cannot meet similarity criteria receive low weights when making Bayesian model average forecasts, there is little to lose from intentionally seeking to create plausible models that represent a stakeholder's overriding concerns. The clear advantage to this approach is that these models, if shown to have low likelihood, can directly address the stakeholder's biases. On the other hand, if these high consequence models are shown to have high likelihood, then the hydrologic analysis has uncovered what may otherwise have become a costly hydrologic surprise following the common single-best-model approach to hydrologic modeling.

Bayesian analysis is often criticized for including an element subjectivity. In particular, Bayesian analysis requires a prior probability estimate, even if data are scarce. The objectively subjective choice may be to assume a uniform prior. But this overweights models that will turn out to be incorrect. Another way to think of this subjective prior is to intentionally allow it to represent the stakeholder's initial conclusions or concerns – perhaps by overweighting the advocacy models. The process of updating as information is added could then be seen as a transparent transition to an increasingly scientifically-informed posterior. Put in the right context, this may be more acceptable than simply asking a stakeholder to replace their model of the world with that of the hydrologist – especially if the hydrologist is honest about their model's shortcomings. From this perspective, capturing subjectivity in the prior (through the inclusion and preferential weighting of advocacy models), far from being a limitation, is critical to providing scientific input that can inform the decision process.

Having delved a bit more deeply into discussing the possible advantages of a Bayesian approach to hydrologic analysis, this final discussion will turn to one example of a more involved application of Bayes' Theorem related to data selection. The preceding discussion was limited to a single observation. In reality, hydrologic models are assessed in the context of multiple observations and observations are planned in sets. The remainder of this discussion considers complications that apply when considering multiple observations simultaneously in a Bayesian context. This discussion is also limited to direct measurements—those that require relatively expensive instruments and/or installation or that face real costs related to data collection or analysis. There are other complications that arise when considering geophysical or other remote sensing data, but those are not addressed here. With that in mind, this discussion will only be relevant as long as direct observations are necessary to characterize subtle processes in complex subsurface environments. By my estimation, that is only likely to apply for the next 100 years.

An observation is valuable if it meets the following four criteria: 1) it contains new information that can be used to assess the relative reliability of models (or to force the development of new models to add to the ensemble); 2) the resulting changes in model probability lead to changes in the probability-weighted decision-relevant forecasts; 3) the resulting changes in forecasts translate to meaningful changes in the Bayesian average utility; and 4) the changes in utility could lead to a change in a stakeholder's decisions. This view is grounded in Bayesian tenets: adding information can modify model likelihoods; but not all information has equal power in this regard. In addition, these statements are guided by decision science: the decision maker translates forecasts into utilities and changes in utility are most important as they approach thresholds that will motivate a change in decision.

Figures 4 and 5 illustrate a Bayesian approach to viewing the value of a single observation. If models forecast values for an observation that are not measurably different, then there is no value in collecting the observation. But when choosing among observations, it is necessary to consider how the evidence provided by different observations translates into changes in the Bayesian model average utility and, in particular, if the observation has the possibility to result in outcomes that cross an important utility threshold for decision making. Simply stated, if every model indicated that a stakeholder should support pumping, and if every plausible measured value of a proposed observation confirmed this conclusion, then there is no potential value in collecting the data. In contrast, if an observation has the possibility to change the ensemble enough to suggest that pumping would cause damage to the stream, then there is value in collecting that observation.

The same Bayesian approach can be applied to a set of observations. The difficulty arises in identifying the composition of the observation set. That is, we could test whether observations O1, O2, and O3 have the possibility of altering a stakeholder's opinion. We can also compare the possible influence of this set to a set comprising O1, O2, and O4. But it is not a simple matter to predict the value of replacing O3 with O4. Specifically, we need to know how much of the information provided by O3 and O4 is independent of (or redundant with) the information in O1 and O2. Further, we need to know how influential the independent information in O3 and O4 are in terms of the ensemble utility with respect to utility thresholds. There may be approaches to approximate this information. But in

the nature of many Bayesian analyses, the following represents a brute force approach that aims to practical use through manageable computational demand.

Strictly speaking, the plots shown in Figure 4 could be replaced with multidimensional relationships that consider the normal distribution over all measurements, including the variance of each observation and the covariances among observations. The covariance among observations could be estimated based on the model forecasts. But this replaces estimates of measurement uncertainty with model variance, which may not be defensible. Instead, we commonly replace the statistical measure of probability illustrated in Figure 4 with a measure of goodness of fit based on the inverse of the sum of squared differences between the forecasts and observations. In the Bayesian context discussed above, this approach is overly optimistic about the value of a near perfect match between a model and data (the limit of goodness of fit is infinite, even if the data are known to have errors). But by summing many observations and relying on the general shortcomings of our models, this approach is usually adequate. The following example uses the inverse root mean square error, normalized by the sum of these values over all models, as the measure of model likelihood.

For this illustration, consider a relatively simple example problem. *A* hydrologic state, *S*, at time, *t*, and distance, *x*, is defined based on the initial state, $S_0$, linear decay with time, exponential decay with time, and an added sinusoidal component. The time of release, *t\**, after time zero varies and is described by the variable, *A*, and there is a constant velocity such that $dx/dt = 5$.

$$S(x,t) = S_0\left((1 - Bt^*) - e^{-Ct^*} + D\left(sin\left(\frac{2\pi t^*}{E}\right) + 1\right)\right) \tag{4}$$

$$t^* = t - A - \frac{dx}{5} \tag{5}$$

The constants (*A*, *B*, *C*, *D*, *E*) are drawn randomly to produce an ensemble of 501 models. The breakthrough curves of *S* as a function of time at the observation point (Figure 6a) and at a more distant point at which a forecast is made (Figure 6b) were calculated with each model. Candidate observations were considered at 10 evenly spaced times (white dashed vertical lines on Figure 6a). Rather than adding random noise to each observation, which requires multiple realizations to avoid the impacts of specific error realizations, each observed value was truncated to the nearest multiple of two times the variance. This approach assumes that measurement uncertainty can be viewed as a loss of measurement precision, but it only requires a single error realization. No noise was added to the model forecasts. The forecast made by each model at the most distant location at time $t = 100$ is subjected to the utility curve (Figure 6c), producing a distribution of utility values over the ensemble (Figure 6d). One example truth model is shown (red lines and points in Figure 6). Note that the concentration of utility values in some ranges is a function of the distribution of forecast values as well as the shape of the utility curve.

A truth model was selected from the ensemble. This model was used to produce simulated observed values, which were truncated to represent the effects of measurement error. The truth model was then removed from the ensemble, leaving a 500-model ensemble. The five best-fit models based on consideration of all 10 observations are shown in green on Figure 6a,b. Note that the models show good agreement with the truth model in the observation period (Figure 6a), but they consistently overpredict *S* at later time (Figure 6b), indicating that the models are not perfectly constrained by the data. In total, 11 truth models were chosen by identifying the models that forecast utilities closest to 0, 0.1 ... 1.0. This was repeated for 10 ensembles of 501 models each. In total, this required 5511 model runs.

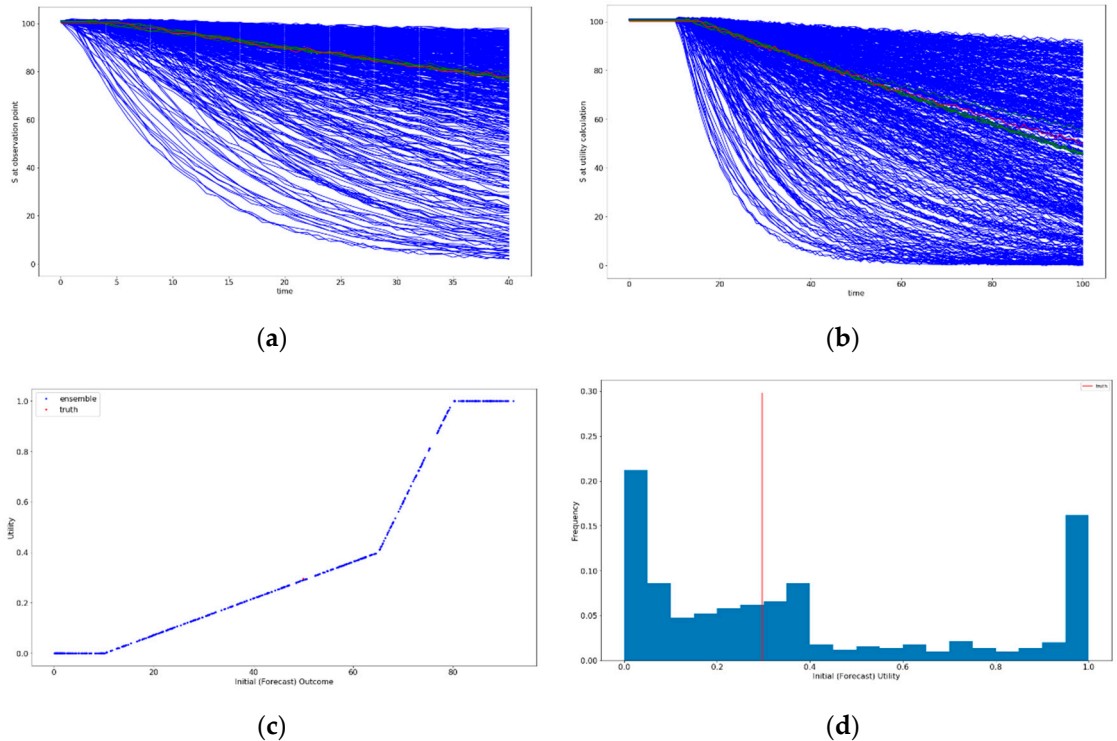

**Figure 6.** Ensemble of model-predicted observations through time with truth model shown in red at (**a**) the observation location and (**b**) the location of the forecast used to calculate utility; (**c**) the utility function; and (**d**) utility probability distribution over the model ensemble.

Each ensemble run produced 10 observations and a forecast for each model. The mismatch from the truth model was calculated for each observation for each model and each value was squared. These are simple matrix calculations, requiring little computational effort. Then, 1025 combinations of the candidate observations were considered, including all possible numbers of observations in the set from 0 to 10. Each observation set was represented as a set of weights: 1 for included observations and 0 for excluded. The product of the mismatch and the weight for each observation was summed over all models, divided by the number of models, and the square root was calculated. This, again, required very little computation effort for each observation set. The computational burden for this analysis could become unmanageable under two conditions: if the model run time is too long to allow for a number of runs equal to the number of realizations multiplied by the size of the ensemble and/or if the number of observations considered becomes too large, given that the number of possible sets rises rapidly with increasing numbers of candidate observations. These limits can be addressed for some cases by limiting the number of realizations and/or the ensemble size or by limiting the number of observations considered or the observation set size allowed. Some models may be too expensive to subject to the following data-worth analysis: but these may also be challenging to calibrate effectively. Finally, the model runs needed for this analysis could be combined with an ongoing ensemble model analysis such that no new model runs are needed specifically for this analysis.

The mismatch for each model was combined to calculate the evidence was for each model for each observation set and the updated model likelihoods were used to provide a Bayesian model average forecast. This was then converted to a utility, leading to an estimated utility for each proposed observation set for each ensemble realization. The performance of a measurement set was based on the mismatch of the Bayesian model average forecast and the truth model's utility. Consider one model ensemble and a selected utility of 0.3. The ensemble forecast utility is shown for each observation set (blue dots, Figure 7a). The average utility calculated for each set size is shown as a red dot. The dashed horizontal red line shows the utility for the truth model. Note that even using all of the

candidate observations, the inferred utility is not identical to that of the truth model; this is due to the divergence of the best fit models from the true model after the observation window. For the case shown, the ensemble forecast utility converges to a consistent value. Any set of five or more observations constrains the utility estimation to this value. Furthermore, there are sets with only one or two observations that result in the same forecast utility value as found using all 10 observations. For each set size, the observations that result in the lowest absolute error compared to using all 10 observations are identified as the optimal set. The compositions of the optimal observation sets are shown in Figure 7b. Comparing these sets across realizations could provide guidance on the optimal composition of sets of different sizes, which could be used as part of a cost:benefit analysis to design an efficient and effective observation network.

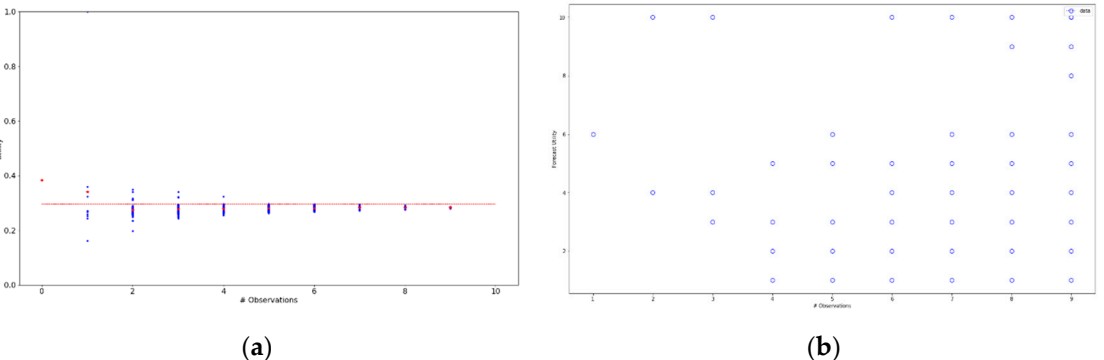

| (a) | (b) |

**Figure 7.** (**a**) Ensemble forecast utility for each observation set for one model ensemble and one selected utility level as a function of the number of observations in the set (blue point for each realization, red star for the average for each number of observations, red dashed line for the true utility); and (**b**) composition of the best set of each size for the sets in panel (**a**).

The preceding analysis identifies an optimal set for each observation set size and it also identifies the average performance as a function of set size. The average performance can be represented as a continuous curve by connecting the red point for each selected utility level for each truth model and each realization. These results (Figure 8a), are color-coded by the utility value used to choose the truth model. The inferred utility value remains stable after two observations for almost all cases. Slightly more data are needed in the steeper regions of the utility curve (between 0.4 and 0.1, on Figure 6c). Using the average utility determined based on three observations is more accurate for lower utility cases (Figure 8b).

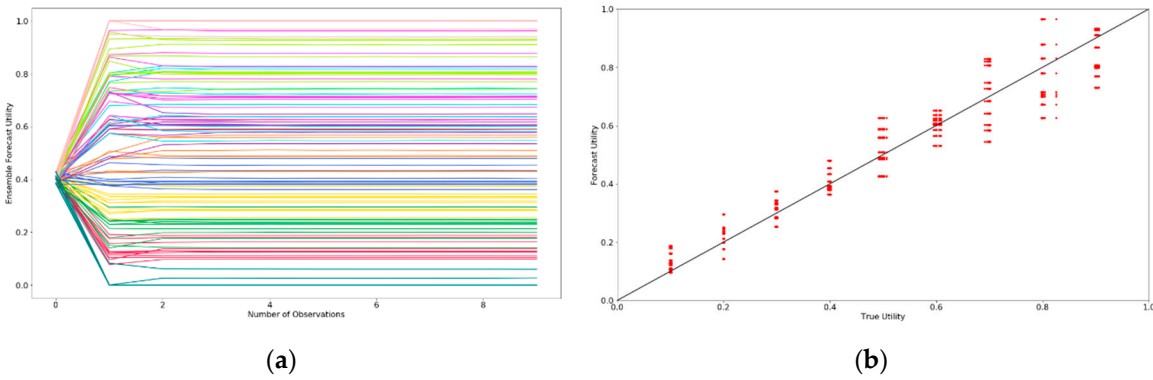

| (a) | (b) |

**Figure 8.** (**a**) Evolution of the average ensemble forecast utility as a function of the number of observations. Colors group analyses by the selected utility of the truth model. (**b**) comparison of the average utility constrained with 3 observations to that of the truth model for all realizations.

The results based on the ensemble average utility, with the model likelihoods determined using the optimal set of three observations for each case, are encouraging. But they are not particularly practical because each realization for each utility level identified a different optimal set (not shown). This does little to help a hydrologist to design a monitoring network. It would be more useful to identify a single optimal set for each set size that is considered over all realizations and truth model utilities. (Alternatively, a set could be derived that places greater weight in the region of the utility values spanning the user-defined threshold for acceptance.) The simplest way to achieve this is to average the performance for each observation set of a given size over all realizations and all utility levels. The optimal set has the lowest average mismatch (highest performance). The progression to a stable utility estimate as a function of observation set size (Figure 9a) shows considerable variability for fewer than four observations. But the inferred utilities become very stable for five or more observations. The composition of the optimal sets (Figure 9b) is fundamentally different than that found through post-audit (Figure 8b). In particular, because a single observation set is found for all conditions, the set builds continuously from smaller to larger sets. That is, once an observation is selected into the set (e.g. for set size 2), those same observations are included for all larger set sizes. Comparison of the utility predicted using this five-observation set, common to all cases, to that predicted by the best three observations, optimized for each realization and utility level, confirms that a single observation set performs well (Figure 9c). Depending upon the application, the added cost of five versus three observations may be more than compensated for by the robustness of the observation set over a wide range of conditions.

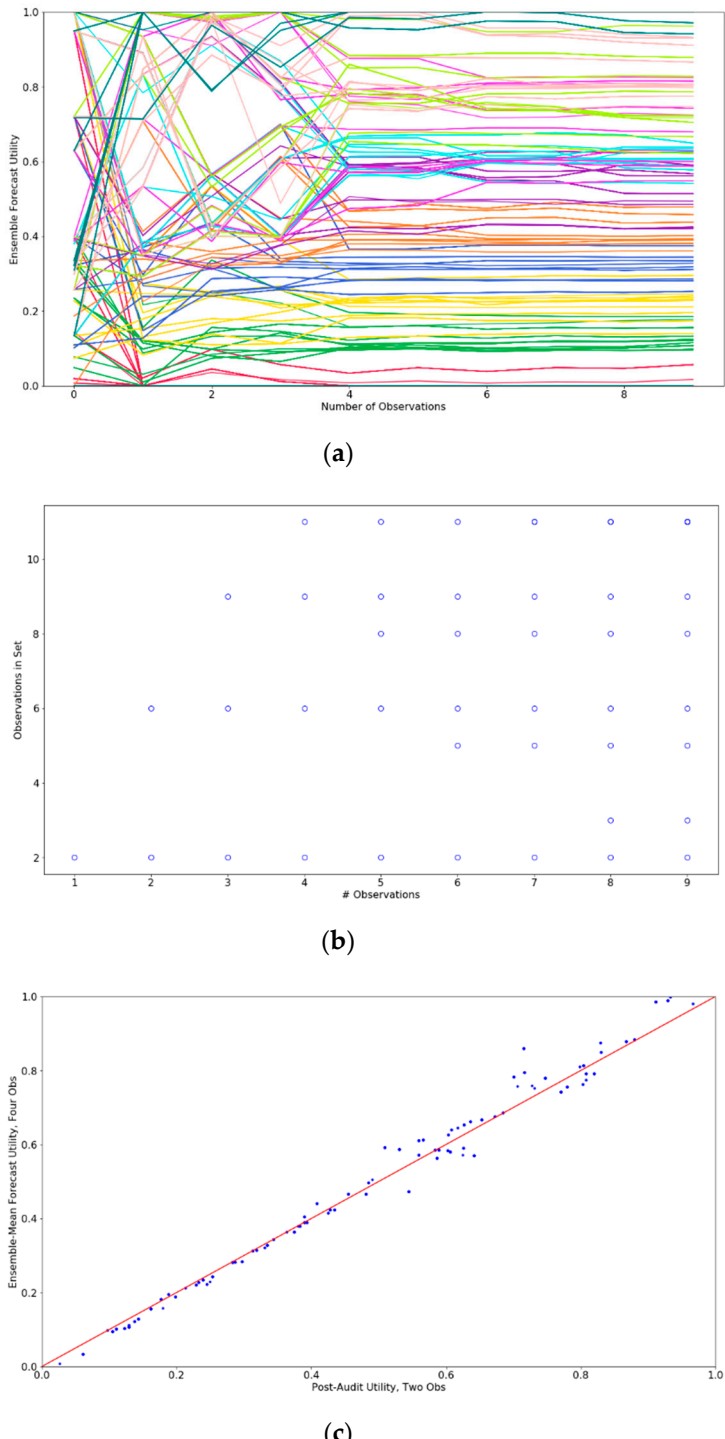

**Figure 9.** Evolution of ensemble forecast utility as a function of the number of observations collected. (**a**) finding the observation that minimizes the utility estimation difference from that determined using all observations for each set size. Colors group analyses by the selected utility of the truth model. (**b**) The composition of the compromise sets. (**c**) comparison of the utility estimated using two observations identified by post-audit and four observations pre-selected using the compromise optimal set approach.

## 5. Summary

This three-part discussion is aimed at three objectives. First, to provide a simple everyday example that could be used to introduce stakeholders to the basic ideas of Bayesian analysis. Second, to provide a more quantitative introduction to Bayesian analysis for beginning hydrologists, including a link

between the stakeholder example and technical terms that are used commonly in this field. Third, to provide one example of the benefits of merging Bayesian analysis and decision theory for more advanced hydrologists. The overall intention is to make the case that we are all, to some degree, Bayesians. We begin with gut-level analyses, minimally informed by data. We seek information that is best able to update our decisions, whether or not it improves our overall understanding of the world. The difference is that scientists should be transparent and as objective as possible in their approach to this updating. One mistake that we make, regardless of our level of experience, is to assume more rigor and objectivity than actually exists in our analyses. We may be well advised to remember these subjective elements of our work and use them to invite subjective input from those who will use our analyses. This begins with the recognition that the stakeholder alone can translate hydrologic forecasts into utilities. It continues with the recognition that we should focus efforts in areas (model development, data collection) that have some possibility to have meaningful impacts on decisions. Finally, embracing some level of subjectivity and outcome-driven science can guide us in the selection of new data for collection that support more objective, science-based updating, while emphasizing reduction of uncertainties that matter to decision makers.

**Funding:** This research received no external funding.

**Acknowledgments:** Too often, reviewing scientific papers is a thankless task. In this case, I want to thank Omar Wani and Akash Koppa for providing extremely useful and constructive critiques of this manuscript. Their insight improved the technical quality of the arguments and fundamentally altered the structure of the presentation. I greatly appreciate their specific, thoughtful comments and positive approach to the review process.

**Conflicts of Interest:** The author declares no conflict of interest.

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
