# Peer review of "Being Bayesian: Discussions from the Perspectives of Stakeholders and Hydrologists"

_water, doi:10.3390/w12020461_

Round 1

Reviewer 1 Report

This manuscript adds to the body of literature on the philosophical basis of Bayesian analysis in the field of hydrology. In part a perspective article, it will contribute to didactics and discussion related to the application of Bayesian analysis in hydrologic sciences. I recommend its publication. 

Comments: 

1) "The goal was to demonstrate that we are all Bayesian in that we make the most of the data that we have, we look for ways to assess the value of those data, and we make projections under uncertainty that guide many of our everyday decisions."

Bayesian updating is a very special case of belief-updating when new data comes in. It is not just any kind of change in belief induced by a confrontation with new data (It has been, for example, shown by Cox's theorem or de Fenitti's constraints on subjective beliefs). Also, Kahneman and Tversky adequately show that the usual heuristics employed by humans are non-Bayesian - with little regard to base rates, beliefs not adding up to the total probability of 1, and proclivity towards more risk-averse behaviors (prospect theory). I would, therefore, recommend reformulating some lines/parts of the first discussion and put proper caveats. Explaining that we all employ different heuristical updating mechanisms in our day to day decision making anyways, and it would serve us well to employ more "coherent" updating mechanisms for more important decision problems, like the ones we face in the field of hydrologic sciences.  

2) A discussion on one of the most important challenges facing Bayesian analysis is missing - i.e. formulation of representative likelihood functions (our models). The likelihood function encodes our knowledge, among other things, on the dependence structure - the one we expect in the spatially and temporally distributed observations. While model selection criteria help us choose one model over the other, there is generally an implicit assumption that at least one of our ensemble models is the "right" observation-generating model, and as we use more and more data for inference, we will be able to identify that model. However, in the case of complex hydrologic systems, we can be sure that we almost never have such a model. A discussion on what such meta-uncertainties (or uncertainty in our uncertainty/probabilistic models) does to a Bayesian analysis would be useful. This paper does a good job of showcasing how Bayesian analysis caters well to some of the very pressing problems in the context of hydrologic decision-making; talking about likelihood functions in somewhat more detail will help the reader to also get familiar with the challenges facing such Bayesian analyses. 

3) Please provide a reformulation of the lines 454 until 467 so that it is more clear what is being presented in figure 6. Including a definition of breakthrough curve. 

Typos and other remarks:

1) Please make the first letter of each sentence capital in the figure captions. 

2) Line 194: models ensemble

3) Line 304: If a several observations

4) Figure 5: remove two full stops. 

5) Figure 7: explanation/legend for the colors is missing

6) line 595: is aimed at three objectives.  

7) Please reconsider the title. "Hydrogeologic" doesn't appear anywhere else in the paper, and "risk" only once. My suggestion (just for inspiration): Being Bayesian: discussions from the perspective of  stakeholders and hydrologists

Author Response

I greatly appreciate your comments.  You pushed me to be much more precise in my definitions and much more thoughtful in the construction of the manuscript.  In fact, the manuscript is largely rewritten following your advice.  In particular, I made a point of starting with an everyday decision for stakeholders to consider rather than beginning with a water resources example.  I used this as a vehicle to make the point that we approach many everyday decisions in an approximately, or qualitatively Bayesian way.  I decided not to go into some of the specific points that you raised about base rates and proofs that Bayesianism is complete.  Rather, I added much more emphasis to the idea that the Bayesian approach is marked by starting with an initial probability distribution estimation that is poorly informed and often subjective and updating that (somewhat informally) as data are collected.  I also wove the important discussion of the incompleteness of the ensemble throughout the manuscript.  This was an excellent suggestion and a truly gaping hole in the first version!  I moved the visual representation of Bayes’ Theorem to the stakeholder section even though it requires a smattering of math.  This allowed me to focus the first hydrologist discussion on a more complete definition of Bayesianism, including reference to some common technical terms that were omitted from the first submission.  Finally, I streamlined the third discussion to focus exclusively on data worth analysis, but to do so with (I believe) a stronger tie to the previous discussions.  To me, the flow is much improved and may actually be readable by each of the three audiences as now written.  Again, I appreciate your guidance!  I also addressed all of your specific comments.  Finally, I changed the title to follow your recommendations as well.

Best

Ty

Reviewer 2 Report

The current manuscript gives interesting explanation on implimentation of Bayesian Theory on hydrolocial application. Discussion is given for three different type of audiences. The first two discussion are more genera and simple, which focus on the decision maker for practice and the general hydrologists whose research topic is not directly related to statiscally analyse on hydrology. The third discussion is provided for specialists who are interested in uncertainty quantification and stochastic hydrology. In general, this paper is a very good overview and introduction paper for different readers who are interested in hydrology, especially the stochastical hydrology. 

I would have two minor comments: 

1) in the introduction section there are too many references, especially in the first two sentences. I doubt that if this is necessary as they are just listed as examples and are not discussed in detail later. I would suggest take a few of them and discuss a bit more details. 

2) for figure 4,5,6,7,8 and 9: in the figure captions a)b)c)... are mentioned but in deed they are missing in the figure. Please add them as figure titles.

Author Response

Thank you for your positive comments and helpful suggestions.  I added the letters to the panels and cleaned up the captions for better correspondence with the figures.  In the end, I decided to keep the references in the introduction section.  I agree entirely that it makes the text less readable.  But, I think that it is useful to point out a wide range of references for the intended audiences.  Thanks again for the positive tone and thoughtful critique!

Best

Ty

Reviewer 3 Report

Review of ‘Being Bayesian: an examination of hydrogeologic risk assessment from two perspectives’

Summary

The manuscript ‘Being Bayesian: an examination of hydrogeologic risk assessment from two perspectives’ by Ferre presents an interesting and much needed discussion on the role of Bayes in decision making in hydrology and water resources, especially on how observational data and models must be used to improve decision making. In doing so, the author makes a convincing argument in favor of being ‘Bayesian’. The author also makes a commendable attempt of improving communication of Bayesian hydrologic analysis to stakeholders who are not hydrologists. The manuscript is well written, and the content is both fascinating and persuasive. However, I have a few reservations about the structure of the manuscript and a few questions on parts of it’s content. In my opinion, the article is suitable, and can be accepted with minor revisions and clarifications, for publication in the special issue ‘Model Uncertainty in Water Science: Conceptualization, Assessment, and Communication’ of the journal Water.

Major Comments

Type of Manuscript: The manuscript has been submitted as an ‘Article’ by which I assume it is to be judged as a ‘Research Article’. However, I think this classification is not entirely accurate as the content of the manuscript is more of an opinion or even a debate. As the author points out in the introduction (line 36) that it is more of a primer or an introduction to Bayesian thinking in hydrology. I suppose that it is up to the editor or the author to clarify this.

Title: I believe that the title does not accurately and adequately represents the contents of the manuscript. The term ‘hydrogeologic’ evokes the picture of groundwater hydrology although the content is more about decision making in hydrology. Also, I am not entirely convinced that the manuscript is concerned with risk assessment. I believe that it is more about decision making and how hydrologic models and data can be used for this in a Bayesian context (Line 41 to 44 distills the essence of the manuscript very effectively in my opinion).

Bayes from a Stakeholder Perspective: Compared to the hydrologist’s perspective, I find the discussion in this section superficial and the structure slightly unintuitive. My suggestion would be to combine compare and contrast how stakeholders and hydrologist would perform the three enumerated steps of ‘being Bayesian’ using the presented example. Currently, the stakeholder perspective enumerates these steps in a linear fashion but in the hydrologist’s approach (Line 392 onwards makes an attempt in this direction) is scattered throughout the subsequent sections (‘Basic Story and ‘Next Level’ sections). The reader would be better served with a condensed comparison of the two perspectives, maybe using a table or a flowchart, and then the further details (single and multiple observations for multimodel hydrologic analysis) can be presented. Also mentioning existing Bayesian techniques  used for multi model ensembles such as Bayesian Model Averaging (BMA) could be mentioned as they are close to the topic discussed in the manuscript.

Bayes from a Hydrologist’s Perspective – the Basic Story: This section is very well written and easy to follow except for the issue detailed above. However, the author avoids using Bayesian terminologies (such as prior, likelihood, posterior etc). In my opinion any hydrologist trying to apply the introduced concepts would be better served if he can relate the contents of the manuscript with the mathematical terminologies he would encounter in other more mathematical texts.

Bayes from a Hydrologist’s Perspective – the Next Level: This section is well written and presents a very convincing theoretical example for analyzing multimodel ensemble forecasts and multiple observations from a Bayesian perspective. However, the applicability of such a method in a real world-case must be discussed. It seems to be that this is a very involved process for deciding the number of observations required to constrain the forecasts. Please comment on the applicability for a real-world case, especially in the age of hydrologic data explosion in the form of remote sensing data.

Minor Comments

Line 114 – ‘…considerable thought instead of ‘…considerable though’

Line 129 – The term ‘model forecasts’ is abruptly introduced at this stage. Maybe some clarification would be necessary.

Line 187 – ‘..analysis is focused’ instead of ‘…analysis focused’.

Author Response

Thank you for your insightful and constructive critique.  The manuscript is considerably revised and greatly improved thanks to your input!  In particular, I have significantly revised the structure of the text.  I have tried to put all of the ‘qualitative’ discussion in the stakeholder section.  This now begins with an everyday decision and tries to lay the groundwork for some of the challenges that face any decision making under uncertainty before introducing a water resources example.  In general, I took to heart your suggestion that the stakeholder section was too superficial, perhaps even bordering on condescending.  It now has a clearer thread and, I believe, makes a stronger connection between everyday experience and a Bayesian hydrologic analysis.  In fact, I think that the stakeholder section could now actually be read by a stakeholder even though it includes a bit of math.  This allowed me to introduce the terms that beginning hydrologists will hear when they are introduced to a Bayesian analysis.  This was a very good suggestion and it was a real missing piece of the initial submission.  Both you and another reviewer pointed out, correctly, that the title didn’t really fit the text: I hope that the revised title is more accurate!  As you suggested, I introduced Bayesian model averaging with the first hydrologist discussion and I think that it tightens the flow of the paper and also introduces this important concept.  I streamlined the final section and, in so doing, found that it was easier to clarify the computational demands of the analyses.  I hope that I now make the case that the required computations are, perhaps surprisingly, modest.  The caveat is that the model has to be relatively fast (as with all Bayesian methods) and you can’t consider a large number of possible observations.  That is spelled out in the text, now.  Finally, I agree entirely that this is more of a comment than a research article.  To be honest, I couldn’t find a suitable category when I submitted; but, I have asked the editor if there is a classification that would be a better fit to the nature of this commentary.

Best

Ty